# Development of Solid-Phase RPA on a Lateral Flow Device for the Detection of Pathogens Related to Sepsis

**DOI:** 10.3390/s20154182

**Published:** 2020-07-28

**Authors:** Alice Jane Heeroma, Christopher Gwenin

**Affiliations:** 1School of Natural Sciences, Bangor University, post code Bangor, Gwynedd, Wales LL57 2UW, UK; chu205@bangor.ac.uk; 2Department of Chemistry, Xi’an Jiaotong-Liverpool University, 111 Ren’ai Road, Suzhou Industrial Park, Suzhou 215123, China

**Keywords:** sepsis, lateral flow immunoassay, solid phase RPA, point-of-care, bacterial infections

## Abstract

Population extended life expectancy has significantly increased the risk of septic shock in an ageing population. Sepsis affects roughly 20 million people every year, resulting in over 11 million deaths. The need for faster more accurate diagnostics and better management is therefore paramount in the fight to prevent these avoidable deaths. Here we report the development of a POC device with the ability to identify a broad range of pathogens on a lateral flow platform. Namely Gram-positive and Gram-negative bacteria. The simple to use laboratory device has the potential to be automated, thus enabling an operator to carry out solid-phase lysis and room temperature RPA in situ, providing accurate results in hours rather than days. Results show there is a potential for a fully automated device in which concepts described in this paper can be integrated into a lateral flow device.

## 1. Introduction

### 1.1. Sepsis

Sepsis has become one of the most abundant reasons for patients being admitted to an intensive care unit (ICU) and for the rapid deterioration of health throughout a hospital stay. Sepsis is caused by an unregulated immune response to localized infection [1,2] and without appropriate treatment this response can advance throughout the body, leading to the immune system mistakenly attacking healthy cells. This aforementioned dysregulation of the immune system can cause a deterioration of organs and tissues, inhibiting responses and ultimately resulting in severe multi-organ failure and death [3]. Annually, in the UK alone it is estimated that there are around 274,000 cases of sepsis [4], with an approximate 28% mortality rate, surpassing lung cancer to become the second biggest cause of death after cardiovascular disease [1]. The rise of sepsis cases was initially believed to be simply the host’s responses to particular pathogens, however, there has been a far more complex interplay between the root cause of the immune response and that of the host body, as explained by Gotts et al. [1]. In addition, the population extended life expectancy creates a demographic that is at high risk of contracting severe infections [5]. In the last 25 years, the population of those over the age of 65 in the UK has risen by 2.2%, resulting in 11.8 million older people who fall into the higher risk category [4]. With age, the immune system experiences a reduced production of essential adaptive immune cells and, consequently, must work harder to fight off an infection, creating a generation of people who are unable to respond to this type of immune challenge [5]. Both the elderly, and young children are susceptible to experience very serve forms of sepsis due to impaired T-cell activity and decreased expression of immune system responses [5], resulting in cases of sepsis among neonates being misdiagnosed [6]. More recently in 2020 the often devastating and long-lasting effects of COVID-19 on the body has increased the host’s susceptibility to sepsis resulting in a significantly increased number of cases. Common indicators of COVID-19 infections include respiratory difficulties, with severe cases being admitted into a hospital for pneumonia and other signs of damage in the majority of organs [7]. Both reports from China and the US have shown links between COVID-19 and potential sepsis cases, with 75% of cases in the Seattle region alone accounting for a suppressed immune response [7] and 28% of cases in China experiencing severe heart damage, which often leads to a 5-fold increase in the risk of death. [8] Patients with a compromised immune system—for instance, those with HIV/AIDS and cancer patients undergoing chemotherapy—are more susceptible to experiencing severe forms of sepsis and hospitalization [9,10]. In addition, cytotoxic chemotherapy causes immunodeficiency in cancer patients and is associated with a much higher risk of bacterial or fungal infection that leads to sepsis [11].

An expansive range of pathogens is often the source of onset sepsis [1]. The ever changing adaption of sepsis also impacts the reported number of cases [12]. The origins of sepsis were initially believed to be evoked by Gram-negative bacteria as a result of cells discharging endotoxins into the bloodstream, which orchestrated a series of events yielding an immune response [13]. However this theory is no longer considered true, with more recent studies [14] showing a wide range of pathogens causing sepsis, with around 47% of cases originating from Gram-positive bacteria, 64% by Gram-negative bacteria and 19% by fungus [15,16]. Viruses have also been shown to cause sepsis, but the number of cases is much smaller, [16] and will therefore not be presented in this study. With this in mind, a wide range of infections can be associated with sepsis, including predominantly pneumonia (64%), bloodstream (15%) and infections in the abdomen, which, if left untreated, can cause them to spread and develop into sepsis [17].

### 1.2. Diagnosis

A variety of different methods were established to determine the presence of sepsis. However, the most common one used in early diagnosis was Systematic Inflammatory Response Syndrome (SIRS) [18,19]. Unfortunately, this method was proved to be too sensitive and would often misdiagnose other illnesses as sepsis. An alternative approach was therefore developed in 2007, known as sequential organ failure assessment score (SOFA) [20]. This point-based system was more specific then SIRS and allowed for an accurate diagnosis [20]. Patients with sepsis typically experience several different ailments and, although SOFA was more accurate then SIRS, it became too complicated and often caused delayed treatment. This was, therefore, modified to quick SOFA (qSOFA) to reduce complexity and make diagnoses easier, allowing for greater use in a resource-limited setting [19]. Once SOFA or qSOFA were preformed, further experiments could be conducted to determine the cause of sepsis [18]. This is routinely carried out with the aid of blood cultures which are considered the “gold standard” [21]. However, blood culture results can take anything between 2–5 days, by which point the patients’ health would have deteriorated dramatically, leading to permanent damage and potential death [22]. Examples of POC devices used today include such methods as Septifast (SF) (Roche Diagnostics, Mannheim, Germany) which has been designed to detect 25 different pathogens relating to sepsis using whole blood samples. Detection takes place within 5 h, considerably faster than blood cultures [22]. Although much faster, a patient’s health can drastically deteriorate over a short period of time, and hence a more rapid approach is often desired. Hence there is a clear need for fast reliable point of care (POC) Sepsis detection.

### 1.3. Lateral Flow

Rapid testing has become key to treating sepsis cases correctly and, as such, a high demand for POC devices has developed over the years [23]. These are near-patient rapid tests that can reduce the time between diagnosis and treatment, are low cost and easily operated by health care workers and potentially people with a lack of background knowledge. These devices also often only require a small sample from patients, making them desirable for neonatal care [24]. Sepsis POC has the potential to improve the time from detection to the treatment of patients, allowing for a reduced death rate and to prevent permanent damage to the organs [25]. Examples of lateral flow being used to detect sepsis can be seen in [26,27], yet many do not target the broad range of pathogens that can lead to sepsis.

Within a nucleic acid lateral flow immunoassay, the signal is detected via a sequence containing a tag, with the most common ones being fluorescence isothiocyanate [28] and biotin, [29] which are complementary to either an antibody bound to the surface of the “test” point or complementary to a tag that is labelled, allowing for visualization [30]. The most familiar method of amplification is polymerase chain reactions; however, more recently isothermal amplification methods, including methods such as recombinase polymerase amplification (RPA) [31] and loop-mediated isothermal amplification (LAMP) [28], have been at the forefront of POC testing [31].

### 1.4. RPA Reactions

Recombinase polymerase amplification (RPA) was first introduced in 2006 by Pipenburg et al. [32] to improve isothermal PCR methods within POC devices. The RPA reactions can circumvent fluctuating temperatures that constrain PCR, running solely at 37 °C [32]. At 37 °C recombinase proteins interact with primers present in the sample mixture, creating a recombinase primer complex that can read across target DNA and bind accordingly, dismantling the hydrogen bonding between the double strand nucleotides and replacing them with complementary regions of the recombinase primer complex [32]. This is then able to amplify accordingly without the aid of fluctuating temperatures to displace adjacent strands [32,33]. Polymerase, similar to that in PCR, is then able to create a new DNA strand that is complementary to the existing DNA. [34] The displaced DNA are stabilized using single-stranded DNA proteins (gp32), typically found in the DNA repair systems of bacteriophages [35,36]. The advantages of RPA reactions are that they are faster and require less energy to get similar results to a PCR reaction with a detection limit of less than 10 copy numbers of DNA [37]. Although there are many advantages of RPA, the reality of such mechanisms shows that there are limitations to this method [38]. Flexibility in the kit formulation to tailor for desired needs is difficult, with only one company, Twist Dx (www.twistdx.co.uk; UK) having the rights to mass-produce such kits [32,38]. In addition, there are issues in PCR and RPA, namely their inability to amplify in the presence of high concentrations of genomic DNA without experiencing a large production of primer artefacts [32,39]. The inability of RPA’s effectiveness in creating amplicons with a base pairing of greater than 300 should also be considered [40]. Notwithstanding this, RPA reactions are one of the most robust and simplest isothermal PCR reactions to use in comparison to other isothermal amplification methods.

Here, we demonstrate the use of RPA reactions on a solid phase surface. The integration of a solid surface reaction, rather than liquid, allows for a more controlled exposure to contaminate and a decreased time to result, which in the future will help to facilitate a potentially fully integrated rapid test as represented in Figure 1. The RPA reactions were performed on cellulose paper and later transferred onto lateral flow devices to be analyzed. Experiments were also conducted to prove that cell lysis could also be performed on cellulose paper, allowing for further studies into the production of a completely integrated solid phase device from sample to result. These methods have the potential to be advantageous in resource limited setting, where the equipment for methods, such as PCR, are not available and the risk of contamination can be higher.

## 2. Materials and Methods

### 2.1. Fabrication of Lateral Flow

First and foremost, the development of the target sequences for the nucleic acid-based detection, are to be considered. Because sepsis is caused by such a broad range of pathogens, sequences that can broadly pick up a specific group of pathogens are ideal. The gene sequence for RPOB [41] concerning Gram-negative bacteria was chosen, MerA [42] for Gram-positive bacteria. Sequences were downloaded from the GenBank^®^database (https://www.ncbi.nlm.nih.gov/GenBank/) and analyzed on BioEdit to create a multiple sequence alignment. Primers were aligned to the guidelines of the TwistAmp^®^DNA amplification kit (TwistDx, Ltd., Maidenhead, UK) and the optimal primer combination was obtained via screening using basic local alignment search tool BLASTN (https://blast.ncbi.nlm.nih.gov/Blast.cgi). All oligonucleotides were synthesized by Eurofins Genomics.

### 2.2. Development of the Solid Phase RPA Reactions

Reagents for RPA reactions were supplied by TwistDx, Ltd., apart from template and primers—the latter were purchased from Eurofins Genomic. In these experiments, nitrocellulose, cellulose and polyvinyl difluoride (PVDF) were initially studied for their ability to retain proteins and other reagents within their pores. Only cellulose was continued onto experiments. To test their compatibility with RPA reactions, experiments were carried out as follows: Cellulose (Whatmann^®^) was cut into thin strips of 20 × 5 mm and saturated with 0.5% bovine serum albumin (BSA, 50 μL) for 30 min. The cellulose was then washed three times for five minutes in nuclease-free water and left to dry at room temperature for 1 h. Next, a master mix of RPA reagents was made up consisting of forwarding primer (1.2 µL, 10 µM), reverse primer (1.2 µL, 10 µM), 12.5 µL of reaction buffer, dNTP’s (4.6 µL, 10 µM) and 2.5 µL of 10 × probe E mix [32], ready to put onto the surface of the solid phase.

Each forward primer has either digoxigenin (DIG) or fluorescein (FAM), whilst each reverse primer contains biotin (BIO). These labels allow for the amplified product to be detected by the antibodies found on the surface of the lateral flow device. A 25 μL solution of “master mix” was pipetted onto the surface of cellulose and left for 30 min to dry [43]. Cellulose is then sealed inside a chamber made from a combination of parafilm and Polytetrafluoroethylene (PTFE). PTFE acted as a sealing tape preventing moister from being lost during a reaction, whilst parafilm was used to prevent the PTFE from directly sticking to the cellulose strip [44]. These prepared strips were then placed at 4 °C until ready for use. The lysis pads were constructed as follows: cellulose was cut into 20 × 5 mm strips. A mixture of (4-(2-hydroxyethyl)-1-piperazineethanesulfonic acid) (HEPES) (1 mM, pH 8), Ethylenediaminetetraacetic acid (EDTA) (500 mM), nonyl phenoxypolyethoxylethanol (NP-40) (10%) and 2,5-Dimethoxy-4-chloro amphetamine (DOC) (10%) were added together in ddH_2_O [45]. Next, 25 μL of this mixture was added to the surface of cellulose and left to dry at room temperature for 1 h. Once dry, the cellulose strips were stored at 4 °C until ready for use.

### 2.3. Lysis of Pathogens

For comparison, each set of experiments were carried out via both solid and liquid phase. For liquid-phase, Escherichia coli (ATCC^®^ 700926™) and bacillus subtilis (ATCC^®^ 6051™) were picked and cultured in 5 mL broth for 16 h at 37 °C. After 16 h, all broths were spun down for 1 min at 10 g to remove liquid and leave the remaining cells. The cells were re-suspended in 200 µL of PBS on ice and then spun for a further 1 min at 37 °C, after which the PBS was removed. Cells were suspended in 200 μL of lysis solution [45], as previously described, and left for 20 min at 37 °C. On completion, the lysis solution was cooled on ice for 10 min and centrifuged for 1 min at 10 g to remove cell debris. The supernatant was removed and 250 µL ice-cold ammonium acetate (7.5 M) was added and held on ice for a further 10 min to precipitate proteins out of solution and centrifuged for 1 min at 10 g and supernatant was poured off into clean Eppendorf [45]. To remove any remaining proteins present, a solution of chloroform:butanol (24:1) was added and gently mixed to separate DNA and proteins into their miscible solutions, causing the aqueous layer (top layer) to contain DNA, whilst the bottom layer contains remaining proteins from the lysed cell [46,47]. Once settled, the aqueous layer was removed and, to this layer, an equivalent of 0.54 to solution ice-cold propan-2-ol was added and inverted gently for 60 s until a white solid was formed. The remaining solution was spun for 20 s at 10 g, producing a pellet of DNA at the bottom of the Eppendorf [46]. The supernatant was removed, and 70% ice-cold ethanol was used to wash the DNA. Once the ethanol had been poured off, the DNA was allowed to dry at room temperature. This was done by partially sealing the Eppendorf with parafilm with holes pierced in it to allow ethanol to escape whilst reducing the chance of contamination. The solid was re-suspended in 200 μL of ddH_2_O and the concentrations of the resulting DNA were confirmed, and the mixture was then diluted in a range between 100–10 ng/μL, depending on how successful extraction was. DNA was then stored at −20 °C until needed.

The solid-phase lysis reactions were similar to the liquid phase, however after pellets were formed and washed with PBS, these were then re-suspended in 500 mM PBS solution and 50 μL was then placed directly onto a pre-made cellulose strip as previously described and impregnated with a lysis solution (25 μL) [44]. The strip was then sealed in parafilm and incubated at 37 °C for 30 min to prevent the surface from drying out. Parafilm was then removed and placed in an Eppendorf and spun for 1 min at 10 g. Cellulose was then removed and a further 50 μL of 500 mM PBS buffer was added to it, surrounded by parafilm, sealed inside two pieces of PTFE and incubated for 10 min at 37 °C. This was done to collect whatever remained on the surface of the cellulose. After 10 min, the cellulose strips were then added to the Eppendorf and spun again for 1 min at 10 g to extract any remaining product from the surface of the cellulose. The cellulose was then removed from the combined solutions of product and the liqueur was treated and the product purified as previously described. The purified product was taken for nanodrop reading to determine the DNA concentration/purity. These methods were also compared with sonication, which was used as a standard.

### 2.4. Primer Design

To establish nucleic acid detection using RPA reactions, the starting point was to identify target gene sequences. Sepsis is caused by a broad range of different pathogens, [48] each with its unique gene sequences. For a simplistic approach, pathogens causing sepsis were split into two categories—Gram-negative bacteria and Gram-positive bacteria and the core gene sequences—established to pick up a broad range of pathogens within each group, whilst avoiding the detection of other pathogens, to increase specificity [49]. For Gram-negative bacteria, the gene sequence RPOB (reference sequence: NC_000913.3) [41], a basic transcription gene was chosen, and for Gram-positive bacteria, mercury reductase (MerA, reference sequence: CP008814.1) [42] was used. Each gene was obtained from the GenBank database (https://www.ncbi.nlm.nih.gov/genbank/) and using BioEdit multiple alignments was performed on each gene sequence comparing it between categories previously described. Primers were designed based on the guidelines provided by the TwistAmp DNA amplification kit (TwistDx Ltd., Maidenhead, UK) and an optimal primer combination was obtained by screening via basic local alignment search tool (https://blast.ncbi.nlm.nih.gov/Blast.cgi) and physical testing. All primers were purchased from Eurofins genomic, as described in Section 2.1.

### 2.5. PCR

All PCR reactions were conducted to prove that the RPA primers designed were able to amplify the desired sections of DNA. The PCR reactions were carried out as follows: to 12.5 μL of superfi master mix (Invitrogen), 1.25 μL of forwarding (10 μM) and 1.25 μL of reverse primer (10 μM) were added and mixed briefly. Following this, 1 μL of 10 ng/μL of DNA was added and again briefly mixed. The mixtures were then placed in a PCR machine and run in a two-step cycle rather than three-step. This was because, although PCR experiments were run, the primers used were RPA primers, and hence were twice the size and did not require specific temperatures [32]. Reaction conditions for amplification were 95 °C for 2 min, followed by cycling conditions of 94 °C for 30 s and 72 °C for 30 s, with a final extension of 72 °C for 5 min. All amplifications were analyzed on 2% agarose gels which ran for 80 min at 50 V. Amplification products were detected using ethidium bromide.

### 2.6. RPA

All materials for RPA reactions, including buffers and reagents, were supplied by TwistDX, UK. Genomic DNA strands were collected from either solid phase or liquid phase chemical lysis, as previously described in Section 2.2. The resulting DNA was diluted to 10 ng/μL with nuclease-free water. Each mixture contained the following: forward primer (1.2 μL, 10 μM), reverse primer (1.2 μL, 10 μM), 12.5 μL of reaction buffer, dNTP’s (4.6 μL, 10 μM) and 2.5 μL of 10 × probe E mix [32]. These were briefly vortexed and, following this, 1.25 μL of 20 × core reaction mix was added and pipetted gently. Next, 10 ng/μL of the template and 1.25 μL of MgOAc were added to the mixture, then vortexed briefly to start the reaction. Samples were then incubated at 37 °C for 20 min. After 4 min, samples were briefly taken out and shaken and returned to the incubator. Samples were then immediately frozen at −80 °C for 10 min. The samples were defrosted ready for use using binding buffer supplied in GeneJET PCR purification kit at a 1:1 ratio and isopropanol on a 0.5:1 ratio [50]. The mixture was pipetted a few times to mix and then added to a purification column supplied from GeneJET. Columns were spun at 10 g for 1 min and the flow-through discarded. Columns were then washed with 100 μL of ice-cold wash buffer (GeneJET) and spun again at 10 g for 1 min with the flow-through discarded. This was repeated twice with a final third spin without any additional wash through to remove any excess wash buffer. Finally, a new clean Eppendorf was used, and the column was added to it. Next, 10 μL of elution buffer (GeneJET) was added to the column and spun for 1 min at 10 g. Resultant products were then either used in lateral flow devices or added to polyacrylamide gels.

The solid-phase reactions used the same materials as the liquid reactions with the cellulose used being prepared the same way as previously described above. However, after the addition, reaction buffer, dNTP’s, 10 × probe E mix, primers and 20 × core reaction mix were added to an Eppendorf and spun down. These were then added to cellulose at –80 °C for 10 min. Following this, they were freeze-dried, allowing for consistent drying along the entire surface of the cellulose [51]. Once fully dry, strips were left at −20 °C until ready for use. Next, 25 μL of nuclease-free water containing template DNA and 1.25 μL of MgOAc was made up in an Eppendorf and mixed briefly. This was then added to the surface of the dried cellulose, which was sealed in parafilm and PTFE and incubated at 37 °C for 30 min. Next, the parafilm and PTFE were removed and cellulose strip was spun for 1 min at 10 g. Cellulose was then soaked with 50 μL of nuclease-free water, left to sit for a further 10 min, and then spun for 10 min at 10 g. The resultant product was then frozen at −80 °C for 10 min. Samples were defrosted with binding buffer and isopropanol in a 1:0.5:1 ratio and spun in a purification column (GeneJET) at 10g for 1 min and the flow-through was discarded. The solution was then washed with ice-cold wash buffer, spun for 1 min at 10g. This was repeated twice and the flow-through discarded. Finally, the column was added to a clean Eppendorf and 10 μL of warm elution buffer was added to the column. These were then spun for 1 min at 10 g and either analyzed via polyacrylamide gel or on the lateral flow device.

Solid-phase lysis was repeated as described above and, following this, the parafilm was removed and lysis paper was pressed onto the surface of the RPA solid phase. These two pieces of cellulose, one from the lysis reaction and one from the RPA reaction, were then sealed together with parafilm and PTFE, as previously described, and left to incubate at 37 °C for 30 min. The resulting product was then extracted off the paper via centrifugation and added to the binding buffer and vortexed briefly. Next, 75 μL of product/buffer was then added to the lateral flow device via the sample pad and left to run for 10 min. The results were observed after 10 min. All experiments were repeated in triplicate.

## 3. Results and Discussion

In addition to the standard technique provided by TwistDx, the RPA reactions were performed via the novel solid-phase technique, as explained in Section 2.4. The results were compared concerning both the sensitivity and time-to-results, as well as cross-reactivity. For all tests, DNA concentrations were prepared from stock solutions that were previously extracted through sonication or chemical lysis, as previously described, and diluted down using nuclease-free water.

### 3.1. Lysis Reactions

All lysis reactions were compared using standard sonication methods along with chemical lysis. This system was designed from a variation of chemical lysis known as radioimmunoprecipitation (RIPA) buffer [45]. The RIPA buffer circumnavigated the demand for a clean-up step between lysis and amplification and ultimately led to a reduced chance of external contamination whilst still being able to produce reliable results. Each sample was placed in 1% agarose gel containing ethidium bromide and analyzed via agarose gel, which can be seen in Figure 2.

The results, as shown in Figure 2, showed that, although lysis was apparent for solid-phase lysis, it was unable to break down as many pathogens, most likely due to the restricted flow of reagents on the solid surface [52]. Results from the nanodrop also indicated a lower yield of DNA extracted via solid-phase lysis, as shown in Figure 3. It should be noted that, when integrating the lysis step with the RPA reactions, however, issues occurred, which are discussed further in Section 3.4.

### 3.2. PCR Reactions

The PCR reactions were initially conducted to confirm that the primer designs were able to bind to genomic DNA extracted from pathogens. The results showed that the RPA primers could be successfully used with PCR reagents (data not shown) allowing for experiments to be continued onto RPA.

### 3.3. Liquid RPA Reactions

Liquid RPA was conducted to prove the effectiveness of the primers designed in RPA reactions and to compare the data sets between liquid and solid-phase results. Although reactions initially worked, analysis via polyacrylamide gels showed that the samples created a bright shadow as they passed through the gel, as well as leaving a larger signal at the very top of the gel. It was concluded that the bright shadow seen at the top of the gels could indicate the remaining recombinase–primer complex present in the sample, thus being too big to pass through the gel [35]. To improve the gels, GeneJET PCR purification kits were used to remove remaining proteins from RPA reactions, allowing for much clearer signals, as shown in Figure 4 [35].

### 3.4. RPA Reactions with Contaminates

The effects of lysis on RPA reactions were determined via “spiking” standard liquid reactions with individual RIPA buffer reagents in addition to a combination of reagents. HEPES (1 M), Na-DOC (10%), EDTA (500 mM) and NP-40 (10%) [45] were each added by substituting the volume of water added to each mixture, making the final concentration of reagents equivalent to that of a standard RIPA reaction. Although HEPES, EDTA and NP-40 showed little effect on RPA reactions, Na-DOC was found to hinder the RPA reactions. The concentration of Na-DOC was therefore adjusted between 2–10%, and initial findings demonstrated that RPA reactions experienced hindrance at a concentration greater than 6%. The literature has also shown that some detergents, such as SDS, also have similar inhibitory effects to amplification [53]. To accommodate the Na-DOC, concentrations were adjusted accordingly.

For completeness, the RPA reactions were also spiked with blood (Thermo Scientific Oxoid SR0050C) to replicate POC use, was used to dilute 10 ng of pathogenic DNA creating a concentration range between 10 and 2 ng. The whole blood samples did not affect amplification, [40] with only a small decrease in amplification seen at 2.5 ng/μL of pathogenic DNA (data not shown).

### 3.5. Solid-Phase RPA Reactions

Initial experiments tested the compatibility of RPA reactions with a solid surface. A master mix, as previously described [32], was made up and pipetted onto the surface of cellulose paper. Primers and template were added together and mixed and pipetted onto the solid surface for 20 min at 37 °C to mimic a standard RPA reaction. Next, 2% BSA was used to prevent none-specific binding; the BSA was coated and dried onto the solid phase before proteins were added. [54] Experiments were repeated in the presence of the blocking agent, which proved valuable, resulting in the efficient removal of proteins bound to the surface [55].

During solid-phase RPA, reagents supplied by TwistDx, dNTP’s and forward/reverse primers, outlined in Section 2.2, were added together in a “master mix” and added to the surface of each solid phase surface and immediately frozen at −80 °C. Following this, the surface was freeze-dried for 10 min. Initially, pads were placed in the freeze dryer and left until completely dry; however, it was noted that as the surface dried the solution concentrated towards the center of the pad, suggesting that there would be an unequal distribution of reagents on the surface. This has been acknowledged before in many papers and is usually provoked by uneven temperature distribution on the surface of the material, creating varying drying times [51,56]. Therefore, the method was adapted, with a brief freezing step at −80 °C to allow for a constant temperature on the surface and allow for even drying speed [50]. Sudden temperature changes also reduced the chances of potential contaminates from interacting with the RPA reagents on the surface, and thus potentially causing a reaction to take place. Once dry, a mixture consisting of DNA, MgOAc and nuclease-free water was pipetted onto the surface, the solid phase was then sealed with parafilm and PTFE. Results were analyzed via either polyacrylamide gels or on the lateral flow devices. Reactions were performed at the same temperature and time to standard liquid reactions; however, at 20 min the signal produced on the polyacrylamide gel was not strong enough to draw any meaningful conclusions, thus the time of 20 min was increased to 30 min to allow for complete use of all reagents in the reaction. This was postulated to be needed to allow for the reactions to take place on the solid surface as, unlike liquid reactions, there is a restricted movement due to the presence of the solid phase. After 30 min, the product was removed via centrifuge and collected in an Eppendorf. The solid phase was then saturated further with nuclease-free water to release any product still left on the surface and centrifuged off and analyzed via polyacrylamide gel or on the lateral flow device, as can be seen in Figure 5.

Figure 5 shows Gram-negative bacteria and Gram-positive bacteria were amplified accordingly, hence both reactions were deemed to be successful. To demonstrate that both solid-phase lysis and RPA could be carried out in succession, lysis was performed on a solid surface. Once 20 min was completed, this “lysis phase” was pressed against a pre-treated RPA solid phase and sealed in PTFE to prevent it from drying out. Products were purified and analyzed via either lateral flow or polyacrylamide gel to prove that the desired reactions had taken place. Results showed that amplification was successful with this method and produced results similar to that of just the solid phase RPA reaction. The set-up also illustrates that the integration of two or more of these reactions may be suitable for integration into a lateral flow (data not shown).

In addition, the reactions were also repeated but without a purification step; however, these were just used on the lateral flow device rather than on a polyacrylamide gel, as proteins did not need to be removed from this as they do not alter the lateral flow results and reduce the likelihood of a reduction in sensitivity from purification. This set of reactions clearly showed, as shown in Figure 5, that the clean-up step is not required for lateral flow devices.

### 3.6. Integration with Lateral Flow Device

All solid-phase work was carried out using lateral flow devices supplied by Abingdon health. Antibodies on these lateral flow devices were complementary to those supplied in the device and allowed for the successful binding between lateral flow device and the product produced. Once solid-phase lysis and RPA were completed, the resulting solutions were extracted from cellulose and added to the running buffer. This mixture of running buffer and RPA reaction was then placed on the sample pad of the lateral flow device and run for 10 min, after which results were photographed, as shown in Figure 6. The results showed that reactions run on solid phases were able to be run on lateral flow devices. Reactions were successful for Gram-negative and Gram-positive primers as illustrated in Figure 6.

With these results in mind, DNA collected from solid-phase lysis reactions were diluted down using nuclease-free water and solid phase RPA reactions were run. Results from these can be seen in Figure 7 with a limit of detection of 2 ng/nL DNA per reaction. A slight shadow can be seen at the top of the gel, which could indicate the presence of a recombinase primer complex still within the sample.

Within Figure 7 the control lane 1 on the gel shows no amplification illustrating that there were no contaminants in the sample. Lane 2 shows amplification as well as artefacts being produced. This can either be from contamination or the RPA reaction amplifying on mismatching complementary strands. Lane 6 is the brightest band due to this having the largest concentration of DNA present. This band also experiences artefacts at approximately 200 bp, which could be due to a large amount of DNA present in the sample, allowing for more potential mismatching and amplification. To conclude, a range of between 100 ng to 0.1 ng/μL was tested which can be seen on both polyacrylamide and lateral flow devices. Anything above can potentially hinder RPA reactions [40].

## 4. Conclusions and Future Work

Accurate and fast POC devices for the detection and correct diagnosis of sepsis will become the forefront in the improvement of the treatment of patients (their subsequent recovery rate) and the battle against the ever increasing number of sepsis cases worldwide. Here, we have described the use of all steps towards the detection of pathogenic DNA via solid-phase lysis, amplification and subsequent detection, which in the future can be developed into a fully integrated device. The ensemble was able to detect various pathogens that commonly induce sepsis, including *E. coli* and *B. subtilis.* In addition to this, RIPA buffer reagents were used to break open the cells and extract the DNA, and these were shown not to affect the RPA reaction—with exception of Na-DOC which demonstrated mild disruption to reactions. Solid-phase RPA reactions were able to detect as little as 2 ng/μL of DNA. This paper also shows the success of a step by step process of lysis followed by RPA, both carried out on a solid phase. However further work would need to be conducted in order to merge methods described with lateral flow devices.

Future work would look into the detection of fungal pathogens that also cause sepsis to create a device that would be able to detect the three major types of pathogen that can induce a septic response. A complete evaluation of the precision and accuracy of the ensemble will also need to be carried out in order to ensure that any arising false positive and negative results will be reduced on a larger sample set. However, in essence, the ensemble shows much promise towards a fully integrated analytical test with the potential to integrate tests done in this paper with lateral flow devices to identify pathogens in a blood sample in under an hour without the need for expensive equipment.

## Figures and Tables

**Figure 1 sensors-20-04182-f001:**
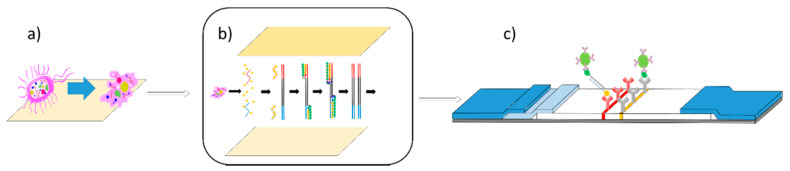
Lab-on-a-chip for solid phase-based lateral flow assay. (**a**) An illustration of solid phase lysis taking place on cellulose paper. (**b**) Illustration of solid phase lysis paper being pressed against solid phase RPA paper, allowing for RPA reactions to then take place. (**c**) Illustration of nucleic acid lateral flow detection of products of solid phase RPA. (Adapted from Piepenburg et al. [32].)

**Figure 2 sensors-20-04182-f002:**
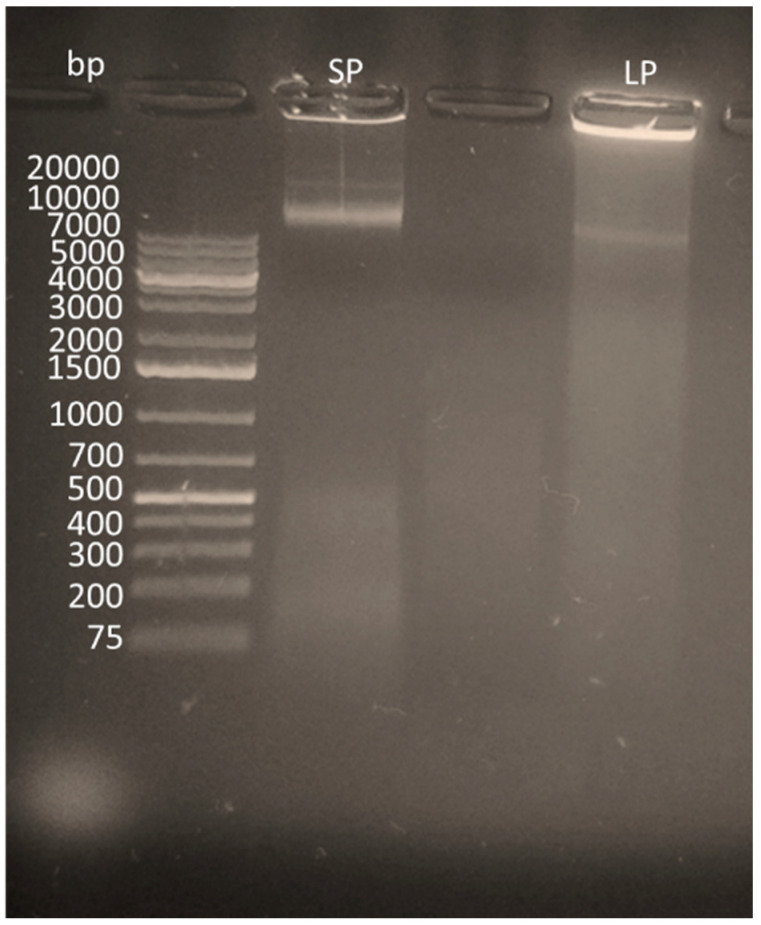
An agarose gel (0.5%) illustrating the lysis on solid-phase (SP) vs. liquid phase (LP).

**Figure 3 sensors-20-04182-f003:**
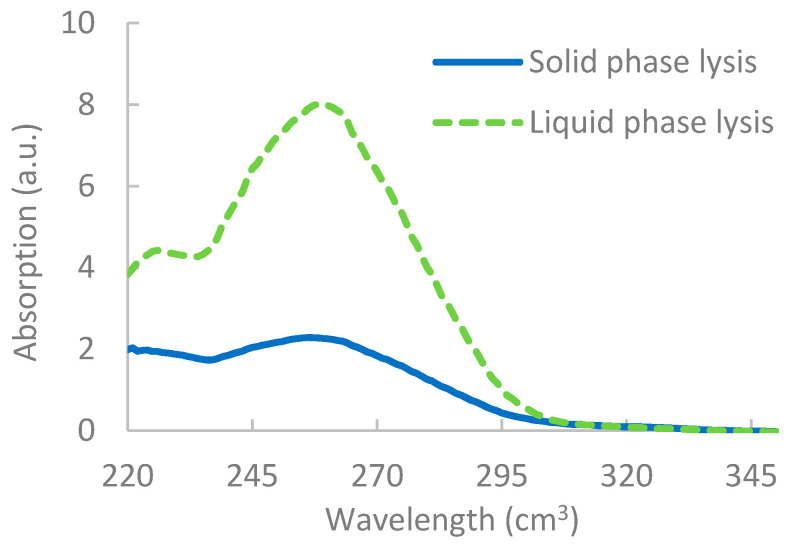
Results from the nanodrop showing the difference in concentration of liquid phase RPA vs. solid-phase RPA. Concentration is determined by measuring the absorption at 260 cm^3^ × dilution factor × 50 ng/mL.

**Figure 4 sensors-20-04182-f004:**
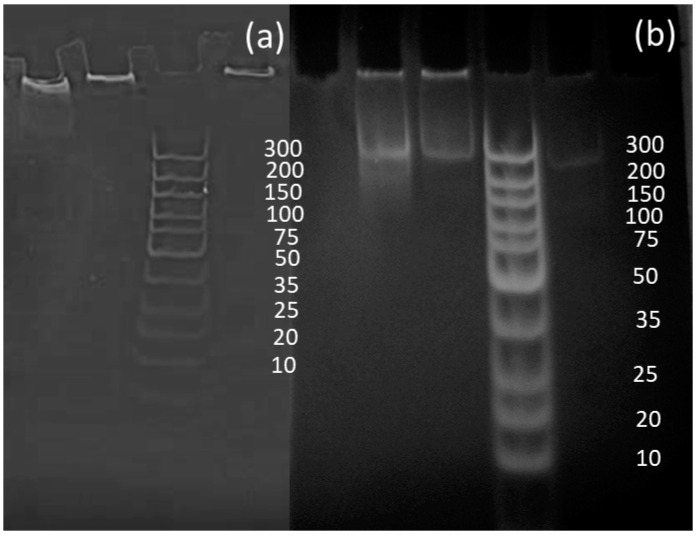
Amplifications of *Escherichia coli* samples (**a**) before purification and (**b**) after purification.

**Figure 5 sensors-20-04182-f005:**
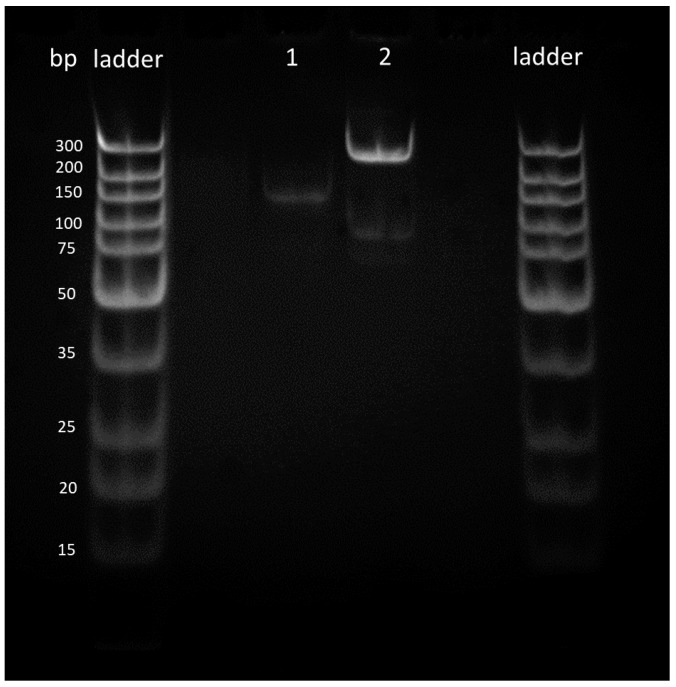
Singleplex solid-phase RPA reactions. Results showed successful amplification of both Gram-negative (2) and Gram-positive bacteria (1).

**Figure 6 sensors-20-04182-f006:**
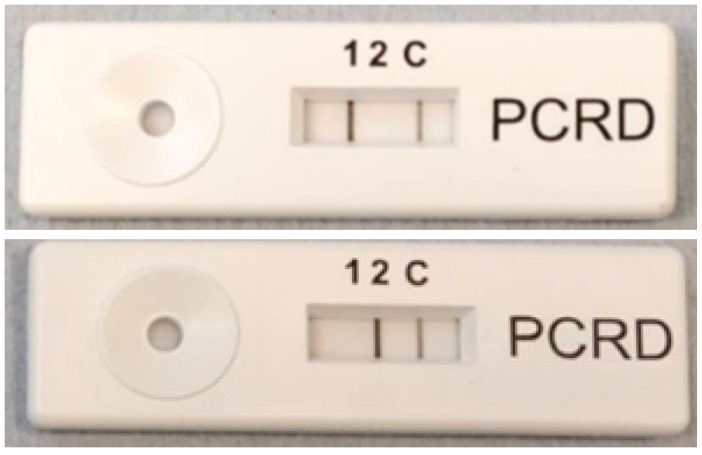
Results from solid-phase lysis followed by solid-phase RPA, analyzed on a lateral flow device supplied by Abingdon health. Line (1) indicates the amplified *B. subtilis* gene whilst line (2) indicates amplified *E. coli* gene. Line (c) is the control line. Top picture shows that *B. subtilis* has been detected whilst bottom picture shows that *E. coli* has been detected.

**Figure 7 sensors-20-04182-f007:**
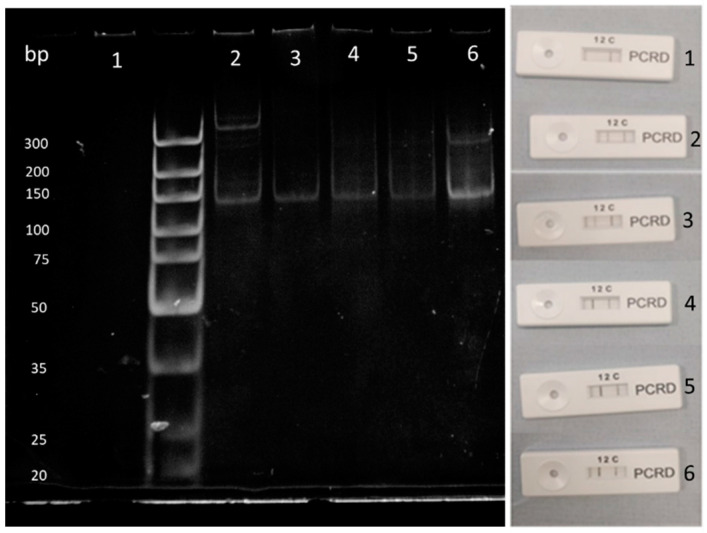
Dilution factors for template DNA of *B. subtilis* from 0 ng/μL (1) to 10 ng/μL (6) compared between lateral flow and polyacrylamide gel. Line (1) indicates the amplified *B. subtilis* gene whilst line (2) indicates amplified *E. coli* gene. Line (C) is the control line.

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
