# Peer review of "Development of Solid-Phase RPA on a Lateral Flow Device for the Detection of Pathogens Related to Sepsis"

_sensors, 2020, doi:10.3390/s20154182_

Round 1

Reviewer 1 Report

Alice Jane Heeroma and Christopher Gwenin they propose a devise with the ability to identify a broad range of pathogens on a lateral flow platform to be able to detect all cases of sepsis in the population and that there are no false positive / negative results and to give timely treatment.

 This is a good work.

The results and conclusions are supported by the methodology and discussion

The title reflects the finding of the manuscript.

The manuscript is for the most part well written and shows robust methods, though some grammatical corrections will be necessary and the use of prepositions should be reviewed. The abstract section is appropriate

They should be homogeneous in the writing of the abbreviations like L,  mL also  the name of bacteria line 173 bacillus subtilis

Although the introduction is appropriate to the study purpose.

Author Response

Comments and Suggestions for Authors

Alice Jane Heeroma and Christopher Gwenin they propose a devise with the ability to identify a broad range of pathogens on a lateral flow platform to be able to detect all cases of sepsis in the population and that there are no false positive / negative results and to give timely treatment.

 This is a good work.

The results and conclusions are supported by the methodology and discussion

The title reflects the finding of the manuscript.

Many thanks

The manuscript is for the most part well written and shows robust methods, though some grammatical corrections will be necessary and the use of prepositions should be reviewed. The abstract section is appropriate

We have now rechecked the use of prepositions and rechecked the grammar – apologies for the errors we hope that we have addressed them all?

They should be homogeneous in the writing of the abbreviations like L,  mL also  the name of bacteria line 173 bacillus subtilis.

We have now made the requested changes many thanks for pointing these out.

Although the introduction is appropriate to the study purpose.

Many thanks

Reviewer 2 Report

The authors present the development of a solid-phase RPA method working in lateral flow to detect pathogens related to sepsis. This work is potentially interesting. However the manuscript fails in making a strong point for the utility of the approach in a real setting, and how this compares with other validated approaches. It should be significantly revised for resubmission. Specifically, critical data showing sensitivity, limit of detection, dynamic range and considerations on specificity and expected accuracy and precision should be added and appropriately discussed. 

Author Response

The authors present the development of a solid-phase RPA method working in lateral flow to detect pathogens related to sepsis. This work is potentially interesting. However, the manuscript fails in making a strong point for the utility of the approach in a real setting, and how this compares with other validated approaches.

The authors apologies and agree with the reviewer thus a paragraph has been added in the abstract and line 146 and line 447 to address this issue. Many thanks for pointing this out.

It should be significantly revised for resubmission. Specifically, critical data showing sensitivity, limit of detection, dynamic range and considerations on specificity and expected accuracy and precision should be added and appropriately discussed. 

We agree and have now added text to reflect this in line 429. With regards to the accuracy and specificity real samples would need to be tested to obtain this thus we have included text at line 436 to reflect this.

Reviewer 3 Report

Comments to the Authors

Title “Development of solid-phase RPA on a lateral flow device for the detection of pathogens related to Sepsis”

This manuscript reported the development of a POC device with the ability to identify a broad range of pathogens on a lateral flow platform. Detailed procedure about how to perform solid-phase lysis, as well as solid-phase RPA nucleic acid amplification has been demonstrated. Although the authors claimed that a POC device has been developed, in fact, the current method with complicated procedure is far away from POC diagnosis.

The concept of the manuscript is to build a platform with paper-based microfluidics for simplified nucleic acid test with isothermal amplification, which could be a promising solution for “sample-in, answer-out” nucleic acid diagnosis with low-cost at resource-poor settings. However, considering the complicated procedure with the described solid-phase lysis or RPA, as well as the reduced efficiency, for example, weakened nucleic acid extraction or increased amplification time, it’s hard to judge what’s the advantages of the proposed method.

Since this study doesn’t show any superiority on nucleic acid analysis comparing with existing methods, for example with liquid-phase one, I’d like to reject its publication.

Author Response

This manuscript reported the development of a POC device with the ability to identify a broad range of pathogens on a lateral flow platform. Detailed procedure about how to perform solid-phase lysis, as well as solid-phase RPA nucleic acid amplification has been demonstrated. Although the authors claimed that a POC device has been developed, in fact, the current method with complicated procedure is far away from POC diagnosis.

We agree with this statement the paper is about the development towards a POC device. Extra points in the abstract and lines 146 and line 447 have now been added to hopefully clarify this point.

The concept of the manuscript is to build a platform with paper-based microfluidics for simplified nucleic acid test with isothermal amplification, which could be a promising solution for “sample-in, answer-out” nucleic acid diagnosis with low-cost at resource-poor settings.

However, considering the complicated procedure with the described solid-phase lysis or RPA, as well as the reduced efficiency, for example, weakened nucleic acid extraction or increased amplification time, it’s hard to judge what’s the advantages of the proposed method.

The advantage is as stated in your first paragraph, however, as this is not coming through in the manuscript extra text has been added to the abstract and lines 146 and line 447.

Since this study doesn’t show any superiority on nucleic acid analysis comparing with existing methods, for example with liquid-phase one, I’d like to reject its publication.

We are hoping that the extra text now clarifies this point it’s the ensemble that is the change in the state of the art.

Round 2

Reviewer 2 Report

The authors addressed my comments and provided a manuscript that can be considered for publication after minor revision. 

As a side note, I believe that the rapidity of the proposed approach represents the strength of the manuscript. This aspect can be further stressed out along the manuscript, particularly in comparison with previous approaches used in the clinical setting. 

Author Response

Many thanks this has now been done

Reviewer 3 Report

Since basically all the comments have been fixed, I'd like to recommend its publication. 

Author Response

many thanks